# Fighting Fire with Fire: Assessing Test Set Contamination Through Deliberate Training on Test Data

## Abstract

Test set contamination poses a serious threat to reliable model evaluation. Whether inadvertent or deliberate, contamination may lead to misrepresenting model capabilities to both researchers and the public: this spurious performance augmentation may in turn cause harm when these models are deployed in real-world applications. In this work, we propose a novel test set contamination detection method that relies solely on analyzing loss trajectories during deliberate fine-tuning on target benchmarks. Our key insight is that models exhibit quantifiably different learning dynamics when exposed to previously encountered versus novel data. Concretely, we systematically fine-tune models on test data mixed within decontaminated data at varying proportions to simulate contamination scenarios, and fine-tune on decontaminated data only to simulate the clean counterparts. We show that clustering methods using as few as 200 data points can distinguish clean from contaminated scenarios with +95% accuracy. Our method also demonstrates superior robustness in detecting contamination of paraphrased evaluation data compared to membership inference attack baselines, which operate at the individual sample level and typically target verbatim matches. Critically, our approach represents a paradigm shift from static detection metrics to dynamic training-based assessment: observing how models react to controlled fine-tuning on target data rather than analyzing fixed outputs or input manipulations. We posit that this intervention-based methodology offers inherently higher resistance to detection evasion, as the metrics cannot be directly optimized as reward signals during model development, providing a more robust foundation for maintaining evaluation integrity.

## 1 Introduction

Test set contamination fundamentally undermines model evaluation by violating the core assumption that evaluation data remains unseen during training. When evaluation benchmarks leak into training corpora—whether through inadvertent web scraping or deliberate inclusion—the resulting performance metrics cease to reflect true model capabilities. This contamination problem has intensified as large language models (LLMs) train on increasingly vast web-scraped datasets that may contain evaluation benchmarks, producing inflated performance reports that mislead researchers and practitioners deploying these models in real-world applications.

Existing contamination detection methods face critical limitations. Static analysis approaches—including output distribution analysis (Yang et al., 2023), perplexity-based metrics (Dong et al., 2024), loss landscape examination, and membership inference attacks (Shokri et al., 2017; Yeom et al., 2018a; Carlini et al., 2022; Shi et al., 2024)—primarily target exact matches and analyze models in their final trained state. This creates two fundamental vulnerabilities: (1) susceptibility to paraphrased or semantically equivalent contamination that preserves meaning while altering surface form, and (2) potential for adversarial evasion, as developers aware of these detection metrics can engineer training procedures to circumvent detection while preserving contamination benefits. Behavioral probes such as the Data Contamination Quiz (Golchin & Surdeanu, 2023) and "Time Travel" guided-instruction prompts (Golchin & Surdeanu, 2024) provide valuable complementary approaches but remain focused on static model responses rather than learning dynamics.

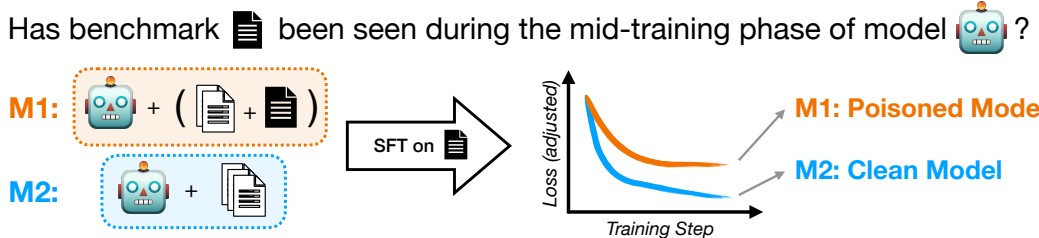

Figure 1: A high-level overview of our procedure. Given a model, we build a classifier that based solely on the loss trajectory's shape–not actual loss values– when fine-tuning on $\mathcal{B}$ it determines whether $\mathcal{B}$ or a dataset similar to it has been seen during training.

We introduce a fundamentally different approach based on analyzing training dynamics rather than static model outputs. Our key insight is that models exhibit measurably distinct learning trajectories when re-exposed to previously encountered versus genuinely novel data during fine-tuning. Specifically, we fine-tune models exclusively on target benchmarks and classify contamination status based solely on the resulting loss trajectory shapes—not absolute loss values, but the temporal patterns of learning.

Our method achieves over 95% detection accuracy across three model families (OLMo-2-0425-1B, Qwen2.5-3B, Qwen3-4B-Base) using Dynamic Time Warping distance metrics with k-nearest neighbors and k-medoids clustering. The approach demonstrates three critical advantages over existing methods: (1) **Cross-model generalization**: 96%+ accuracy when training classifiers on one model and evaluating on others, including effective 1B-to-4B parameter transfer; (2) **Robustness to semantic variations**: maintaining detection performance on paraphrased evaluation data where traditional membership inference attacks typically fail; (3) **Early detection capability**: achieving 95%+ accuracy using only the first 16 trajectory points, corresponding to exposure to just 256 evaluation samples.

Beyond binary detection, our approach enables contamination severity quantification through regression on spurious performance gains, achieving Mean Absolute Error of 0.15 when predicting the magnitude of contamination-induced performance improvements. Even in challenging scenarios involving partial contamination of unknown benchmark subsets, detection accuracy exceeds 80%.

Our approach represents a paradigm shift toward intervention-based evaluation integrity assessment. By observing how models react to controlled training interventions rather than analyzing static outputs, we create detection mechanisms that are inherently more difficult to game during model development. The training dynamics we analyze cannot be directly optimized as reward signals, providing a more robust foundation for maintaining evaluation integrity in an era where benchmark performance increasingly drives development decisions compared to exchangeability tests (Oren et al., 2023) or static distributional analyses.

These findings suggest that training dynamics contain rich information about a model's prior experiences, opening new avenues for understanding model behavior and detecting other forms of training irregularities beyond contamination detection. As models increasingly train on vast, web-scraped datasets where benchmark leakage becomes unavoidable, robust contamination detection becomes critical for ensuring reliable assessment of true model capabilities.

## 2 TEST SET CONTAMINATION DETECTION FRAMEWORK

### 2.1 EXPERIMENTAL DESIGN

**Setup** Let $\mathcal{D}$ be a general, large-scale supervised fine-tuning (SFT) dataset. To detect test set contamination through loss trajectory analysis, we simulate contamination scenarios by training models either on a target benchmark $\mathcal{B}$ mixed with general training data $\mathcal{D}' \subseteq \mathcal{D}$ (a *poisoned* model), or solely on general training data (a *clean* model). We assume $\mathcal{B} \cap \mathcal{D} = \emptyset$.

Let the *poisoning ratio* $p \in (0, 1]$ represent the fraction of a poisoned model's total training data that consists of benchmark samples: $p = |\mathcal{B}|/(|\mathcal{B}| + |\mathcal{D}|)$. Let $M^{(0)}$ be a base model. We define:

- $M_{\text{poison}}^{(1)}(\mathcal{B}, p)$: model $M^{(0)}$ fine-tuned on $\mathcal{B} \cup \mathcal{D}'$ with poisoning ratio $p$. To ensure poisoning ratio $p$, we set $|\mathcal{D}'| := |\mathcal{B}|/p - |\mathcal{B}|$.

- $M_{\text{clean}}^{(1)}(|\mathcal{B}|, p)$: model $M^{(0)}$ fine-tuned on $\mathcal{D}'' \subseteq \mathcal{D}$ only, where $|\mathcal{D}''| = |\mathcal{B}|/p$.

This ensures both models are trained on the same total number of samples while maintaining the target poisoning ratio for the contaminated model. Then, the difference in benchmark performance between a poisoned model and its corresponding clean model corresponds to the *spurious performance augmentation* $\Delta := \text{Perf}(M_{\text{poison}}^{(1)}, \mathcal{B}) - \text{Perf}(M_{\text{clean}}^{(1)}, \mathcal{B})$.

**Loss Trajectory as the Sole Training Signal**  We use the loss trajectory obtained from fine-tuning trained models solely on the target benchmark as our classification signal. Given a trained model $M^{(1)}$ (either clean or poisoned), we fine-tune it exclusively on benchmark $\mathcal{B}$ and record the resulting *loss trajectory* $\ell(M^{(1)}, \mathcal{B}) = (\ell_1, \ell_2, \ldots, \ell_T)$, where $\ell_t$ represents the training loss on the minibatch from $\mathcal{B}$ used at step $t$. This curve $\ell(M^{(1)}, \mathcal{B})$ serves as the sole feature vector for our contamination detection methods.

**Problem Formulation**  Our primary setting is binary classification: given several trained models $M^{(1)}$ and their corresponding loss trajectories $\ell(M^{(1)}, \mathcal{B})$, we classify each model as either clean or poisoned. To ensure meaningful labels, we only retain poisoned models with spurious performance augmentation $\Delta > \delta$ for some fixed threshold $\delta \geq 0$. In addition, we also consider a regression version of this task where we directly predict the continuous contamination strength $\Delta$ providing fine-grained contamination severity estimates rather than binary labels.

## 2.2 Modeling to Distinguish and Quantify Poisoned and Clean Settings

Our modeling choices aim to build classifiers in low-resource settings, and retain interpretability when possible to gain insights into training dynamics. Our problem setup is naturally low-resource since obtaining a single loss trajectory implies fine-tuning an LLM for two epochs.

We use two classification approaches: (1) k-NN classification and (2) k-medoids clustering, both using Dynamic Time Warping (DTW) distance. k-NN performs classification via majority vote of k nearest neighbors, while k-medoids clustering uses the Partitioning Around Medoids (PAM) algorithm to select actual data points as cluster representatives (*medoids*), offering added interpretability through concrete exemplar loss patterns that characterize clean versus contaminated dynamics. For the regression version of our task, we adapt k-NN to predict continuous poisoning ratios via weighted averaging.

**Distance Metric between Loss Trajectories**  Given the sequential nature of loss trajectories, we employ Dynamic Time Warping (DTW) (Sakoe & Chiba, 2003) as our distance metric. DTW computes the minimum cumulative cost to align two sequences by allowing elastic matching—each point in one sequence can match to multiple consecutive points in another while preserving temporal order. For trajectories $\ell^{(i)} = (\ell_1^{(i)}, \ldots, \ell_{T_i}^{(i)})$ and $\ell^{(j)} = (\ell_1^{(j)}, \ldots, \ell_{T_j}^{(j)})$, $\text{DTW}(\ell^{(i)}, \ell^{(j)}) := \min_\pi \sum_{(s,t) \in \pi} |\ell_s^{(i)} - \ell_t^{(j)}|$ where $\pi$ is a warping path—a sequence of index pairs $(s, t)$ that aligns elements from both trajectories while satisfying monotonicity and boundary constraints. DTW is computed via dynamic programming as: $D(s, t) = (|\ell_s^{(i)} - \ell_t^{(j)}|)^2 + \min\{D(s-1, t), D(s, t-1), D(s-1, t-1)\}$ with boundary conditions $D(1, 1) = (|\ell_1^{(i)} - \ell_1^{(j)}|)^2$ and $D(s, 0) = D(0, t) = \infty$. The DTW distance is $\text{DTW}(\ell^{(i)}, \ell^{(j)}) = \sqrt{D(T_i, T_j)}$. This dynamic programming approach allows each point in one trajectory to align with multiple points in the other while preserving monotonicity, making it robust to temporal variations in loss dynamics.

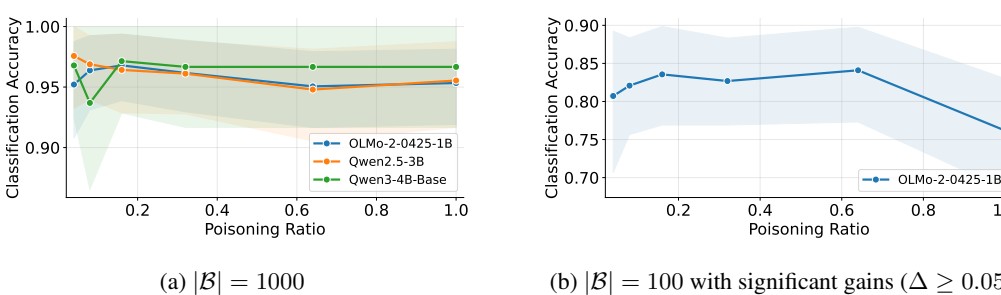

(a) $|\mathcal{B}| = 1000$          (b) $|\mathcal{B}| = 100$ with significant gains ($\Delta \geq 0.05$)

Figure 2: Classification accuracy for detecting whether a model has been fully poisoned with $\mathcal{B}$ at poisoning ratios $p \in \{0.04, 0.08, 0.16, 0.32, 0.64, 1\}$ using 8NN+DTW classifiers for each (model, poisoning ratio) configuration. 20-fold Cross-validation partitioning by benchmarks.

## 3 SPURIOUS PERFORMANCE AUGMENTATION DETECTION

**Experimental Setup** We perform supervised fine-tuning (SFT) taking the loss signal from both the prompts and completions, emulating the standard mid-training setup. We use 38 benchmarks with at least 1000 samples in lm-eval-harness (Gao et al., 2024), consisting of either of multiple choice questions, or are evaluated as short-form text generation where there are several possible options and the one with lowest perplexity is taken as correct, and cover a variety of domains (see App. A). Thus, all benchmarks are evaluated with some form of accuracy score $\in [0, 1]$, where higher is better. We use OLMo-2-0425-1B (**?**), Qwen2.5-3B (Qwen et al., 2025), and Qwen3-4B-Base (Yang et al., 2025) for our experiments. We prioritized models that have not been post-trained for our SFT experiments (the Qwen models) to imitate the usual LLM training pipeline, and included a model from a different family to measure generalizability. We train $\mathcal{M}^{(0)}$ models with two variants (learning rates 1e-6 and 1e-5; 1 epoch) to create $\mathcal{M}^{(1)}$ models. For loss trajectory generation, we fine-tune $\mathcal{M}^{(1)}$ models with a single variant (learning rate 1e-5; 1 epoch) given that is our intervention to the models $\mathcal{M}^{(1)}$ and not a simulation of data poisoning. For loss trajectories, we subsample uniformly the loss trajectory to retain only 64 steps. Each step consists of 8 samples (our batch size). We collect a total of $\approx 4500$ curves, adding up all clean and poisoned models, across model families and poisoning ratios. See App A.

**Baselines** Traditional Membership Inference Attacks (MIA) methods classify individual data samples, while our task requires classifying models as clean or contaminated. To provide fair comparison, we adapt three MIA baselines by computing per-sample scores across the benchmark and treating the resulting score vectors as features for model classification. Since these are not time-series, we use 8NN classification with Manhattan distance instead of DTW: (1) *Loss*: Per-sample loss values on the target benchmark (Yeom et al., 2018b), (2) *Zlib*: Compression ratios for each sample (Carlini et al., 2021), and (3) *Min-K*: Minimum probability among k% lowest-probability tokens (Shi et al., 2024).

### 3.1 DETECTING SPURIOUS PERFORMANCE AUGMENTATION ACROSS MODELS AND BENCHMARKS

**Strong Detection Across Benchmarks and Poisoning Ratios** Our method effectively detects contamination across varying poisoning ratios and model architectures. Figure 2 demonstrates strong performance with classification accuracy consistently above 95% when $|\mathcal{B}| = 1000$ and substantial performance even with smaller benchmarks ($|\mathcal{B}| = 100$), where some models may not achieve meaningful gains from one-epoch fine-tuning. We therefore restrict $|\mathcal{B}| = 100$ analysis to cases with $\Delta \geq 0.05$ to focus on detectable contamination effects (see App. A.1 for performance gains distribution plots). We perform 20-fold cross validation across benchmarks to prevent leakage. Accuracy decreases at higher poisoning ratios for OLMo2-1B, potentially due to limited gradient updates making contamination harder to distinguish. Table 1 validates strong cross-model generalization, with classifiers trained jointly on all poisoning ratios exhibiting robust performance across models.

Table 1: Contamination detection performance using 8NN+DTW with 20-fold cross-validation partitioning by benchmarks. Results show mean accuracy $\pm$ standard deviation across poisoning ratios.

| Model used in training | Performance |
|---|---|
| OLMo-2-0425-1B | $0.958 \pm 0.105$ |
| Qwen2.5-3B | $0.937 \pm 0.127$ |
| Qwen3-4B-Base | $0.966 \pm 0.099$ |

Table 2: Cross-model contamination detection performance using 2NN+DTW. Classifiers trained on one model and evaluated on another, with results showing accuracy across all poisoning ratios and benchmarks.

| Model used in training | Model used for evaluation | | |
|---|---|---|---|
| | OLMo2 1B | Qwen2.5 3B | Qwen3 4B Base |
| OLMo-2-0425-1B | – | 0.971 | 0.979 |
| Qwen2.5-3B | 0.978 | – | 0.998 |
| Qwen3-4B-Base | 0.967 | 0.967 | – |

**Strong Cross-Model Detection, Across Model Families and Including 1B-to-4B Transfer** Table 2 shows that our method works across different model families, with cross-model detection accuracy consistently above 96%. Notably, training on smaller models and evaluating on larger ones maintains strong performance (e.g., OLMo2-1B $\rightarrow$ Qwen3-4B: 0.979), which is practically useful since smaller models require less computational overhead for training classifiers. The consistent bidirectional performance suggests that contamination patterns in loss trajectories generalize across model scales and families.

### 3.2 EARLY TRAINING DYNAMICS RELIABLY REVEAL CONTAMINATION STATUS

Beyond high performance, our method provides some insight into the training dynamics that aid in classification. Figure 3a demonstrates that the 8-medoids clustering achieves strong performance (acc=0.956, AUC=0.98) while producing interpretable groupings: medoids represent actual loss trajectories that characterize each cluster, with clean medoids (M1, M3, M4, M7, M8) distinctly exhibiting sharp initial loss drops, while contaminated medoids show more gradual decay patterns. The high cluster purity indicates that clusters meaningfully separate contamination behaviors rather than relying on spurious correlations. Moreover, Figure 3b reveals that these discriminative patterns emerge early during fine-tuning: classification accuracy reaches over 95% within just 16 loss data points (corresponding to $16 \times 16$ benchmark samples) and plateaus thereafter.

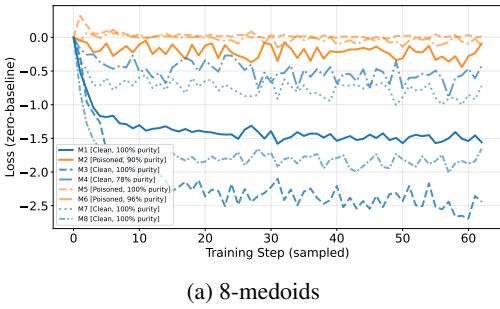

(a) 8-medoids

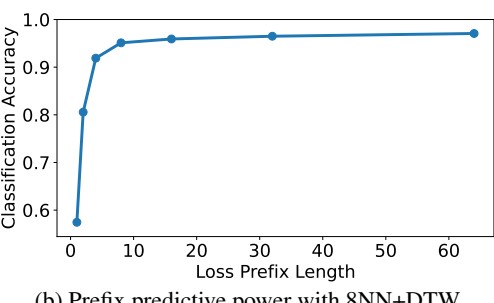

(b) Prefix predictive power with 8NN+DTW

Figure 3: Insights into distinct contamination patterns in loss trajectories. (a) 8-medoids clustering of loss trajectories by contamination status, with class purity indicating percentage of cluster elements matching the medoid's true label. (b) Classification accuracy using 8NN+DTW as a function of loss trajectory prefix length. Results from training on OLMo2-1B and evaluating on Qwen2.5-3B ($|\mathcal{B}| = 1000$, loss data points subsampled as described in Experimental Setup).

### 3.3 DETECTION REMAINS EFFECTIVE WITH NON-VERBATIM TEST SET CONTAMINATION

Real-world contamination may involve semantically equivalent rather than verbatim test data, making detection more challenging for traditional methods. To evaluate robustness against such scenarios, we poison models with benchmark $\mathcal{B}$ and assess detection performance using loss trajectories from either the original $\mathcal{B}$ or paraphrased data $\widetilde{\mathcal{B}}$ (generated with GPT-4o). Figure 4 shows both

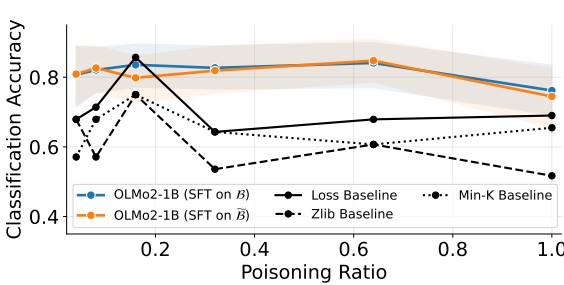 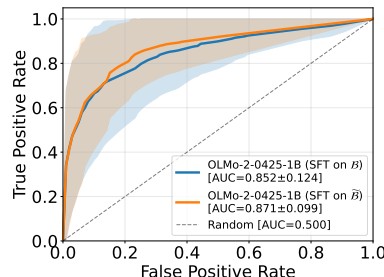

(a) Classification accuracy training each poisoning ratio separately, with 20-fold cross validation.

(b) ROC curve considering all poisoning ratios jointly.

Figure 4: Detection accuracy when leaking a benchmark $\mathcal{B}$ with $|\mathcal{B}| = 100$ and $\Delta \geq 0.05$ (requires minimally meaningful performance gains), comparing when collecting loss trajectories from $\mathcal{B}$ or from a semantically-equivalent dataset $\widetilde{\mathcal{B}}$. 8NN+DTW classifier. See $\Delta \geq 0$ in Figure 9.

test sets achieve identical detection accuracy across poisoning ratios for $|\mathcal{B}| = 100$ when restricting to test set contaminations that lead to meaningful performance gains ($\Delta \geq 0.05$; Figure 4a); some classification accuracy is lost when considering cases with little gains, which we include for completeness but does not represent our target behavior (see Figure 9).

While the most realistic setup would be to poison with paraphrased data and evaluate on both versions, this inverse setup demonstrates our method captures learning dynamics that persist despite textual differences, providing robustness against sophisticated contamination while requiring half the computation since $M^{(1)}$ models were already computed for other experiments.

**Baseline Comparison (MIA)** Figure 4a shows our method consistently outperforms MIA baselines across poisoning ratios. While MIA methods achieve 0.6-0.7 accuracy, our approach maintains 0.8+ performance even for small benchmarks ($|B| = 100$). These results demonstrates that dynamic loss trajectories provide richer contamination signals than static sample-level indicators.

## 3.4 DETECTION UNDER PARTIAL TEST SET CONTAMINATION SCENARIOS

In practice, contamination may involve only a subset of the full benchmark rather than the entire test set. To evaluate detection under these conditions, we assess performance when models are contaminated with an unknown subset $\mathcal{B}' \subset \mathcal{B}$ where $|\mathcal{B}'| = 100$ from benchmarks with $|\mathcal{B}| = 1000$. Figure 5 compares detection accuracy when fine-tuning on the leaked subset versus the full benchmark, showing $> 80\%$ accuracy on average across poisoning ratios, and demonstrating applicability to partial contamination scenarios. Counterintuitively, performance is even stronger when fine-tuning using the whole $\mathcal{B}$ rather than having access to the smaller leaked subset $\mathcal{B}'$—we hypothesize this is because of having access to a larger training set for loss trajectory generation, as Figure 3b already showed increased predictive power for longer curves.

Our method's consistent performance on both original $\mathcal{B}$ and paraphrased $\widetilde{\mathcal{B}}$ data demonstrates robustness against semantic contamination—a key advantage over methods relying on exact string matching. The learning dynamics we capture reflect deeper patterns that persist despite surface textual variations. The $\Delta \geq 0.05$ threshold filters cases with negligible performance gains that would not represent meaningful contamination in practice.

## 3.5 PREDICTING SPURIOUS PERFORMANCE GAINS BEYOND BINARY CLASSIFICATION

Our methodology enables extending beyond binary contamination detection to predict the magnitude of spurious performance gains $\Delta$, where clean models have $\Delta = 0$ and contaminated models may have $\Delta > 0$. Using loss trajectories as the only features, we train 8NN+DTW regressors with Mean Absolute Error (MAE; the absolute difference between predicted and true $\Delta$) as the evaluation metric. Table 3 shows MAE of $\approx 0.15$ when predicting only on contaminated models—a more challenging setting than including all models, since clean model detection is already highly reliable

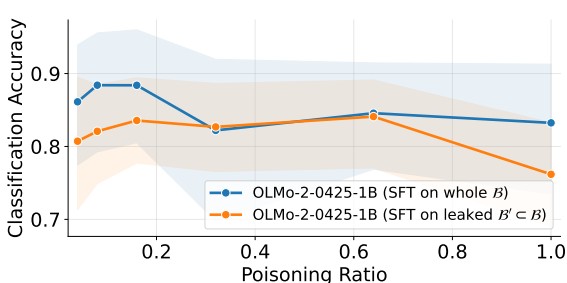 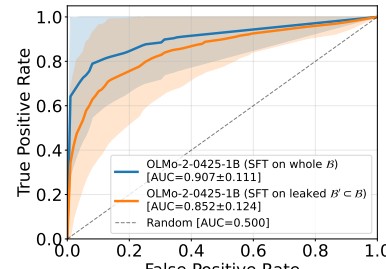

(a) Classification accuracy training each poisoning ratio separately, with 20-fold cross validation.

(b) ROC curve considering all poisoning ratios jointly.

Figure 5: Detection accuracy when leaking an unknown subset $\mathcal{B}' \subset \mathcal{B}$ with $|\mathcal{B}'| = 100$ from benchmarks $\mathcal{B}$ with $|\mathcal{B}| = 1000$, compared to the case where we perfectly know the leaked set. We use 8NN+DTW, over all cases with $\Delta \geq 0.05$.

Table 3: Regression performance for predicting spurious performance gains $\Delta$ using 8NN+DTW on $|\mathcal{B}| = 1000$ settings. Mean Absolute Error (MAE; lower is better) shown as mean $\pm$ standard deviation across 20-fold cross-validation folds split across benchmarks. "Poisoned" includes only contaminated models; "All" includes both clean and contaminated models.

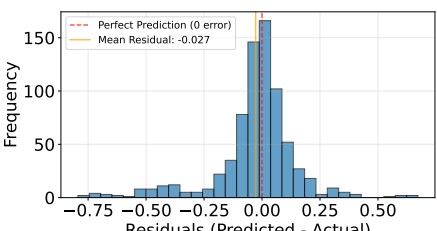

Figure 6: Distribution of prediction residuals (predicted $\Delta$ - actual $\Delta$) for contaminated models only. 8NN+DTW regressor trained on OLMo2-1B trajectories and evaluated on Qwen2.5-3B using 8NN+DTW. All results in App A.2.

| Model | MAE (poisoned) | MAE (all) |
|---|---|---|
| OLMo-2-0425-1B | $0.157 \pm 0.086$ | $0.073 \pm 0.034$ |
| Qwen2.5-3B | $0.147 \pm 0.067$ | $0.070 \pm 0.037$ |
| Qwen3-4B-Base | $0.154 \pm 0.089$ | $0.063 \pm 0.040$ |

for $|\mathcal{B}| = 1000$. Figure 6 shows residuals (predicted $\Delta$ minus actual $\Delta$) with mean of -0.027 and substantial variance, indicating that loss trajectories contain useful signal but perfect prediction remains elusive. Appendix A.2 shows extensive cross-model evaluation showing MAE 0.11-0.19 for contaminated-only settings and MAE of just 0.05-0.09 when including clean models. This regression formulation opens research directions for quantifying contamination impact rather than solely detecting test set contamination, with the loss trajectory as a useful first signal.

## 4 RELATED WORK

**Test Set Contamination Detection**   Related to our work is research that detects benchmark leakage via black-box statistical tests or static behavioral probes rather than training dynamics. Oren et al. test exchangeability by comparing canonical vs. shuffled orderings to provide provable, low–false-positive evidence of contamination in black-box LMs (Oren et al., 2023). Behavioral probes include the Data Contamination Quiz (multiple-choice perturbations) and "Time Travel" guided-instruction prompts that elicit memorized continuations from dataset names and leading spans (Golchin & Surdeanu, 2023; 2024). Survey and benchmarking efforts map contamination at dataset/instance/sub-instance levels and synthesize mitigation directions (Xu et al., 2024), while perplexity- and distribution-based detectors (e.g., CDD/TED) operationalize output-distribution peakedness to flag leakage (Yang et al., 2023; Dong et al., 2024). In contrast, we intervene on the model by *fine-tuning on the target benchmark* and classify contamination from the *shape of the loss trajectory* (rather than one-shot likelihoods or order preferences), enabling robust detection—including under paraphrases—by clustering/nearest-neighboring entire loss curves. This dy-

namic, training-based audit aligns with our central claim that previously seen vs. novel test data induce measurably different learning dynamics during continued training.

**Membership Inference Attacks**   Related to our work are MIAs that infer per-example membership from static model behavior. Classical and reference-based attacks include confidence/loss thresholding (Shokri et al., 2017; Yeom et al., 2018a), LiRA's likelihood-ratio test against shadow references (Carlini et al., 2022), and robust likelihood-ratio testing (RMIA) with improved power at very low FPRs (Zarifzadeh et al., 2024). For LLMs, neighborhood comparison replaces reference datasets with synthetic neighbors (Mattern et al., 2023); Min- k% and Min-k%k++ leverage the tail of token probabilities and a theoretically motivated local-maxima criterion to detect pretraining data (Shi et al., 2024; Zhang et al., 2024). Semantic MIAs train a classifier over meaning-preserving perturbations to improve robustness to superficial lexical changes (Mozaffari & Marathe, 2024), while a large-scale study finds many MIAs approach random guessing on realistic LLM pretraining regimes absent distribution shift (Duan et al., 2024). Closer to our signal are works using *trajectories*: in vision, loss trajectories across intermediate models improve membership detection (Liu et al., 2022); and contemporaneous work studies recovering/approximating training data from weights or training dynamics (Morris et al., 2025). Distinct from these, we treat the *entire loss-time curve during deliberate fine-tuning on the suspect benchmark* as the feature—eschewing static, one-pass scores and reference models—and show strong separability of clean vs. poisoned regimes via DTW-based clustering/nearest neighbors.

**RL, SFT and Memorization**   Related to our work is research dissecting how post-training choices affect memorization vs. generalization. In RLHF pipelines for code completion, RLHF tends to *reduce* memorization relative to direct fine-tuning, though samples memorized during SFT often remain memorized after RL (Pappu et al., 2024). Yet RL/RM stages can exploit spurious signals: response-length bias explains a surprisingly large fraction of RLHF gains (Singhal et al., 2023), and broader analyses document reward hacking/misgeneralization (Bu et al., 2025). In parallel, training-dynamics studies show token-level learning/forgetting signatures that are stable across runs (Chang et al., 2024), while pretraining work charts how larger models memorize faster and forget less (Tirumala et al., 2022). Very recent audits argue that dramatic RL gains on certain math benchmarks (e.g., Qwen2.5-Math) can be confounded by contamination, with improvements vanishing on leakage-free data (Wu et al., 2025). Our approach is complementary: rather than proposing a new post-training algorithm, we *audit* SFT/RL models by probing how their loss decays when trained on target benchmarks—seeking contamination fingerprints in training dynamics instead of static outputs.

## 5   DISCUSSION

We introduce a contamination detection method that analyzes training dynamics rather than static outputs, achieving over 95% accuracy by classifying loss trajectory shapes during fine-tuning on target benchmarks. The method generalizes across model architectures (96% cross-model accuracy), maintains performance on paraphrased evaluation data, and detects partial contamination with >80% accuracy using only 16 trajectory points. Unlike static detection metrics that can be optimized during training, our intervention-based approach analyzes emergent learning dynamics that resist manipulation. Our task formulation and method also enable contamination severity quantification through regression on spurious performance gains, allowing a fine-grained assessment beyond binary classification. As models increasingly train on web-scraped data where benchmark leakage is unavoidable, robust contamination detection becomes critical for evaluation integrity. Our findings demonstrate that models retain detectable signatures of their training history in learning dynamics, providing a foundation for trustworthy evaluation.

**Limitations and Future Work**   Our evaluation assumes specific contamination scenarios that may not encompass all real-world practices. Adversaries might use lower poisoning ratios, sophisticated paraphrasing techniques, or semantically similar data that differ from our LLM-generated paraphrases. Additionally, post-training procedures like RLHF could alter the learning dynamics we rely on, potentially affecting detection robustness. Investigating detection effectiveness across diverse contamination strategies and post-training regimes, particularly reinforcement learning pipelines, represents important directions for strengthening contamination auditing capabilities.

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

## A  ADDITIONAL EXPERIMENTAL DETAILS

**Benchmarks** Throughout this paper we used 38 benchmarks, all hosted in lm-eval. They focus on MCQ or short form generation benchmarks, and we list them exhaustively: `anli_r1`, `anli_r2`, `anli_r3`, `arc_easy`, `arc_eu_easy`, `bigbench_bbq_lite_json_multiple_choice`, `bigbench_color_multiple_choice`, `bigbench_cs_algorithms_multiple_choice`, `bigbench_fact_checker_multiple_choice`, `bigbench_hyperbaton_multiple_choice`, `bigbench_navigate_multiple_choice`, `bigbench_social_iqa_multiple_choice`, `bigbench_temporal_sequences_multiple_choice`, `bigbench_winowhy_multiple_choice`, `boolq`, `eus_exams_es_osakidetza6c`, `eus_trivia`, `inverse_scaling_pattern_matching_suppression`, `mastermind_24_easy`, `mastermind_35_hard`, `mastermind_46_easy`, `mastermind_46_hard`, `medmcqa`, `medqa_4options`, `mnli`, `mnli_mismatch`, `moral_stories`, `multirc`, `persona_optionality-increasing`, `persona_optionality-preservation`, `persona_politically-conservative`, `persona_politically-liberal`, `persona_resource-acquisition`, `prost`, `qnli`, `swag`, `sycophancy_on_philpapers2020`, `wmdp_bio`.

### A.1  DISTRIBUTIONAL DETAILS ON CLEAN AND POISONED MODELS

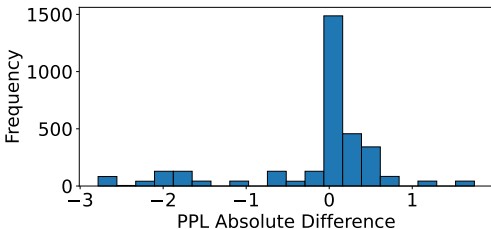

Figure 7: Histogram of perplexity (PPL) changes between the original model $M^{(0)}$ and $M^{(1)}_{\text{clean}}$ models, measured with the `wikitext` dataset. A lack of a significant absolute perplexity difference serves as a validation of preserved modeling capabilities.

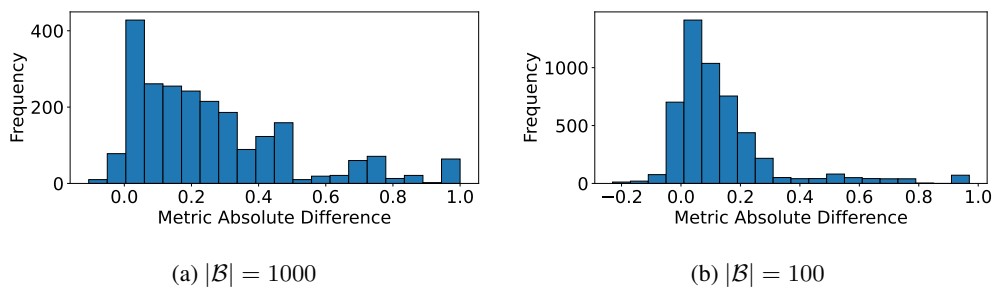

(a) $|\mathcal{B}| = 1000$        (b) $|\mathcal{B}| = 100$

Figure 8: Histogram of $\Delta$ values depending on benchmark $\mathcal{B}$ size. Naturally, training on a larger benchmark generally implies a much larger performance augmentation. Interestingly, when training on a smaller benchmark, there is often no significant performance augmentation.

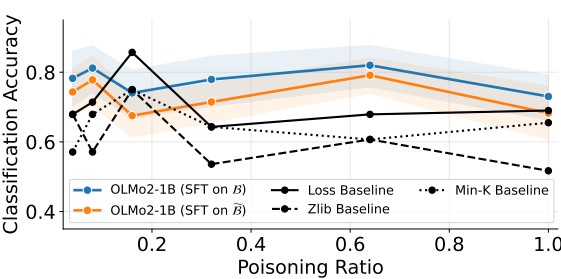 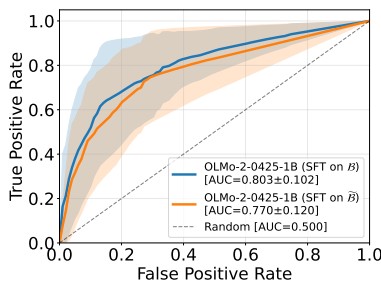

(a) Classification accuracy training each poisoning ratio separately, with 20-fold cross validation.

(b) ROC curve considering all poisoning ratios jointly.

Figure 9: Detection accuracy when leaking a benchmark $\mathcal{B}$ with $|\mathcal{B}| = 100$ and $\Delta \geq 0$, comparing when collecting loss trajectories from $\mathcal{B}$ or from a semantically-equivalent dataset $\widetilde{\mathcal{B}}$. 8NN+DTW classifier. Extension of Figure 4 with $\Delta \geq 0$ (does not require meaningful gains).

## A.2 BEYOND BINARY CLASSIFICATION

Table 4: Regression performance for predicting spurious performance gains $\Delta$ using 8NN+DTW cross-model. Regressors trained on one model and evaluated on another, with Mean Absolute Error (MAE) used as metric to measure distance between the predicted and true $\Delta$, as visualized in Figure 6. "Poisoned" includes only contaminated models; "All" includes both clean and contaminated models.

| Model used in training | Evaluation Models (MAE; poisoned only) | | | Evaluation Models (MAE; all) | | |
|---|---|---|---|---|---|---|
| | OLMo2 1B | Qwen2.5 3B | Qwen3 4B Base | OLMo2 1B | Qwen2.5 3B | Qwen3 4B Base |
| OLMo-2-0425-1B | – | 0.112 | 0.166 | – | 0.052 | 0.063 |
| Qwen2.5-3B | 0.105 | – | 0.145 | 0.047 | – | 0.053 |
| Qwen3-4B-Base | 0.189 | 0.162 | – | 0.086 | 0.071 | – |

