# OpenReview forum: "Fighting Fire with Fire: Assessing Test Set Contamination Through Deliberate Training on Test Data"
_ICLR.cc/2026/Conference — Submitted to ICLR 2026_

### Official Review · Reviewer_tHiN · 2025-10-27

**Soundness:** 3
**Presentation:** 1
**Contribution:** 3
**Rating:** 4
**Confidence:** 3

**Summary:**

This paper proposed a novel detection method for the test set contamination problem, by distinguishing the training dynamic curves on test data of clean and cheated models.

**Strengths:**

1. The proposed detection method is novel in the sense that it depends on the dynamics of further training on targeted data, rather than on static metrics.
2. The detection accuracy is high.
3. It's robust to paraphrasing.
4. It derives a performance gain prediction tool.

**Weaknesses:**

1. The largest model evaluated is of 4B size. If the proposed method is not able to scale up to larger models, the contribution would be limited.
2. The authors should clarify the knowledge and capability of the contamination detector via a formal threat model. For example, in your setting, the detector has access to the model weights, which is a strong assumption. I think only open-sourced models give this while-box access to the detector. Moreover, the detector also knows the training algorithm (SFT and its details), which is not practical in real-world scenarios.
3. In practice, different models usually use different training algorithms and hyperparameters. Thus, the ideal results on cross-model generalization cannot support real-world practice: training a classifier with a fixed model and using it to discriminate any new model.
4. The experimental details are so ambiguous that I can not confidently identify the pros and cons of the reported results. Especially, the authors need to clarify which curves are used for training the classifier in each figure and table.

    - For example, in Figure 2 (a), at the dot of (Qwen 2.5-3B, 0.16), what does the curve set for the classifier contain? All curves with various model archs and various poisoning ratios? Or only curves with  Qwen 2.5-3B, and clean and 0.16 poisoned cases?
    - Similarly, what are the training and evaluation settings for Table 1? Lines 214-215 say that it's about cross-model generalization. But it seems that the model used in training is the same one for evaluation.
    - Similarly, in Table 2, for each row, how do you collect curves and train the classifier? For each model arch, do you collect curves with various poisoning ratios and train only one classifier? Or do you train different classifiers for different evaluation ratios?

**Questions:**

See weakness.

---

### Official Review · Reviewer_Bx44 · 2025-10-31

**Soundness:** 2
**Presentation:** 2
**Contribution:** 2
**Rating:** 2
**Confidence:** 4

**Summary:**

The paper tackles the critical problem of test set contamination in LLMs and proposes an intervention-based contamination detection method that leverages the analysis of loss trajectories from deliberately fine-tuning models on the test set. By comparing the dynamics of loss decay, the method distinguishes between models trained with and without access to benchmark data. The dynamic, training-based audit is shown to outperform the MIA baseline.

**Strengths:**

1. The paper introduces a compelling departure from static, output-based contamination detection by focusing instead on loss trajectory dynamics induced by deliberate fine-tuning, which is interesting to me.
2. Accurate contamination detection can be achieved using only a fraction of the loss trajectory, enabling early identification with as few as 256 samples or the first 16 data points (Fig. 3b).

**Weaknesses:**

1. Firstly, I have to say the novelty is limited, as some of the previous works have already discussed using the loss trajectory for MIA, despite being cited. The conceptual novelty is therefore slightly less pronounced outside the context of LLMs, and the authors would strengthen their case by discussing distinct challenges in the LLM versus vision or other modalities.
2. The proposed Dynamic Time Warping (DTW) distance between loss trajectories seems to be intuitive and lacks formal justification or theoretical analysis connecting observed loss dynamics with contamination levels, and no bounds or convergence results are discussed regarding the reliability of such signals.
3. The manuscript’s writing and presentation could benefit from further refinement. For instance, the full names and abbreviations of several terms (e.g., DTW and MIA) are repeated multiple times; the related-work section is not always expressed in a formal academic tone; a few equations are typeset in a way that may strike readers as informal; several sentences still carry noticeable traces of LLM generation; and there are minor typographical issues (e.g., line 184: “We use OLMo-2-0425-1B (?)”; line 193: “See App A.”).
4. Figure 1 may be confusing because the trend sketched in the diagram does not fully align with the textual description. Replacing the schematic illustration with the actual experimental curves could clarify the point.
5. The evaluation covers only a small set of datasets and models, and the chosen models are neither very recent nor fully representative of the current state of the art.
6. The paper does not include ablation studies that would explicitly demonstrate the effectiveness of the proposed method.
7. I am somewhat surprised to see the paper end with a “Discussion" and "Limitation” section rather than a “Conclusion.” A concise concluding summary would help readers better appreciate the key takeaways.

**Questions:**

1. How does the proposed contamination detection method generalize to benchmarks involving free-form generation, regression tasks, or highly unbalanced labels?
2. Is there empirical evidence or theory to support how post-training stages might affect the detectability of contamination in loss trajectories, especially in practice?
3. What are the computational costs of the proposed method?

---

### Official Review · Reviewer_4auC · 2025-10-31

**Soundness:** 3
**Presentation:** 3
**Contribution:** 2
**Rating:** 4
**Confidence:** 3

**Summary:**

The paper proposes an intervention-based audit for test-set contamination: deliberate fine-tune a model on the target benchmark itself. The method is implemented through recording the loss trajectory during fine-tuning, and classify whether the model had prior exposure to the benchmark by comparing the shape of that trajectory (via DTW + kNN/k-medoids). The study spans three model families (OLMo-2, Qwen2.5, Qwen3) and benchmarks. For the experimental results, the authors report >95% accuracy for clean vs. contaminated detection, and early detection performance. They also show the ability to estimate the magnitude of spurious gains.

**Strengths:**

1. **Simple but effective, intervention-centric approach.** Different from prior methods, shifting the signal from static outputs to training dynamics is refreshing. The focus on trajectory shape instead of absolute loss value fits this motivation.
2. **Broad quantitative sweep.** The experimental grid is reasonably large (three model families; dozens of benchmarks) with consistent gains in detection acc comparing with those using static outputs signals.
3. **Early-signal detection.** The experimetns show that ≈256 samples already enough for high accuracy makes the method more practical.

**Weaknesses:**

1. **Realism & scope of contamination.** The paper simulates contamination by SFT on mixed data with poisoning ratios p rather than during pre-training. It’s plausible that pre-training or RLHF leaves different fingerprints than SFT exposure. The authors acknowledge some threats (e.g., RLHF altering dynamics), but the empirical settings are still simplifying too much.
2. **Practicality for third-party models auditing.** The approach requires fine-tuning the suspect model on the test benchmark to collect loss trajectories. That’s feasible for open models, but many real audits target closed models that either cannot access parameters or cannot be trained for IP/policy reasons. Even for open models, the paper’s approach still implies non-trivial compute.
3. **Sensitivity to training recipe.** The classifier is keyed to optimization traces: optimizer, batch size, lr, number of steps , and the exact SFT objective. The paper varies LR modestly, a deeper robustness slice may help.
4.  **Baselines feel underpowered. ** The adapted baselines are per-sample MIA scores aggregated and then fed to 8-NNclassifiers. This may undersell stronger MIA pipelines (e.g., LiRA[1]/RMIA-like[2] likelihood ratio testing). Stronger baseline choices would strengthen claims.

[1] Carlini, Nicholas, et al. "Membership inference attacks from first principles." SP, 2022.
[2] Zarifzadeh, Sajjad, Philippe Liu, and Reza Shokri. "Low-cost high-power membership inference attacks." ICML, 2024.

**Questions:**

1. **Closed-model audit.** If the suspect model cannot be fine-tuned (typical for frontier APIs), what’s your recommended adaptation to this  scenarios?
2. **Training-recipe invariance.** How sensitive is detection to optimizer type choice, batch size, LR schedule (cosine vs. constant), fine tuning steps?
3. **Compute & time overhead.** What is the end-to-end cost per audited model/benchmark (wall-clock, GPU hours, memory)?

---

### Meta-Review · Area_Chair_GYw9 · 2026-01-02

**Summary:**

The paper proposed a method to detect whether model is trained on a clean dataset or a dataset with contaminations from the test dataset. The method retrains the model on the test dataset and makes the detection by comparing the training dynamic curves. The method is new to certain extent and outperforms the MIA baseline in the setting of the paper (SFT training details and white-box).

The main limitations include:
1. Limited novelty. As mentioned in the paper (line 46), loss landscape examination was considered before, and the authors should mentioned the differences in LLM setting.
2. Limited practicability. The method works for open source models and the training details (optimizer, batch size, lr, et al) of SFT need also known.
3. Limited coverage. Only SFT is considered; RLHF is not considered.

**Reviewer Concerns:**

The following concern can be addressed
1. Limited novelty. As mentioned in the paper (line 46), loss landscape examination was considered before, and the authors should mentioned the differences in LLM setting.

The following concerns are essential
1. Limited practicability. The method works for open source models and the training details (optimizer, batch size, lr, et al) of SFT need also known.
2. Limited coverage. Only SFT is considered; RLHF is not considered.

**Reviewer Scores:**

Reviewer Bx44: The score of Reviewer Bx44 is bit low can can be increased.

---

### Decision · Program_Chairs · 2026-01-26

Reject